# Hospitalizations and Clinical Outcome in Metastatic Colorectal Cancer During Regorafenib or TAS-102 Therapy

**DOI:** 10.3390/cancers12102812

**Published:** 2020-09-30

**Authors:** Florian Huemer, Gudrun Piringer, Verena Schlintl, Hubert Hackl, Gabriel Rinnerthaler, Josef Thaler, Richard Greil, Lukas Weiss

**Affiliations:** 1Department of Internal Medicine III with Haematology, Medical Oncology, Haemostaseology, Infectiology and Rheumatology, Oncologic Center, Salzburg Cancer Research Institute—Laboratory for Immunological and Molecular Cancer Research (SCRI-LIMCR), Center for Clinical Cancer and Immunology Trials (CCCIT), Paracelsus Medical University, 5020 Salzburg, Austria; f.huemer@salk.at (F.H.); v.schlintl@salk.at (V.S.); g.rinnerthaler@salk.at (G.R.); r.greil@salk.at (R.G.); 2Department of Internal Medicine IV, Hematology and Oncology, Klinikum Wels-Grieskirchen, 4600 Wels, Austria; gudrun.piringer@klinikum-wegr.at (G.P.); josef.thaler@klinikum-wegr.at (J.T.); 3Johannes Kepler University Linz, 4040 Linz, Austria; 4Division of Bioinformatics, Biocenter, Medical University of Innsbruck, 6020 Innsbruck, Austria; hubert.hackl@i-med.ac.at; 5Cancer Cluster Salzburg, 5020 Salzburg, Austria

**Keywords:** hospitalization, regorafenib, trifluridine/tipiracil, TAS-102, mCRC, survival, third-line, fourth-line

## Abstract

**Simple Summary:**

Regorafenib and TAS-102 showed a survival benefit against placebo, and both drugs are approved for the treatment of metastatic colorectal cancer (mCRC) beyond second-line. The highly differential toxicity profile of both substances has led to a potentially biased perception of drug tolerability and complications—such as hospitalization—in the oncologic community. The aim of this retrospective analysis was to investigate hospitalization frequency during regorafenib and TAS-102 treatment and the impact of hospitalizations on survival. Treatment with regorafenib as well as a low Eastern Cooperative Oncology Group (ECOG) performance status turned out to be independent risk factors for hospitalization. Hospitalizations due to gastrointestinal toxicity were only seen with regorafenib. However, hospitalizations during regorafenib or TAS-102 treatment did not impact survival. In light of increased gastrointestinal toxicity leading to hospitalization during regorafenib treatment, we call for increased awareness to drug-specific toxicities, in order to prevent unnecessary complications by the early detection of adverse events and prompt counteraction.

**Abstract:**

Current National Comprehensive Cancer Network (NCCN) and European Society of Medical Oncology (ESMO) guidelines recommend regorafenib or trifluridine/tipiracil (TAS-102) for the third-line therapy of metastatic colorectal cancer (mCRC). In this analysis, we evaluated hospitalizations during regorafenib or TAS-102 treatment and the impact of hospitalizations on overall survival (OS). This retrospective analysis was based on unselected, consecutive mCRC patients treated with regorafenib and/or TAS-102 at the tertiary cancer centers in Salzburg and Wels-Grieskirchen, Austria. Between January 2013 and May 2019, 93 patients started third- or fourth-line therapy with regorafenib or TAS-102. Tumor therapy (regorafenib versus TAS-102, HR: 1.95 [95% CI: 1.07–3.54], *p* = 0.03) and the Eastern Cooperative Oncology Group (ECOG) performance status (2–3 versus 0–1, HR: 4.04 [95% CI: 2.11–7.71], *p* < 0.001) showed a statistically significant association with hospitalization risk in multivariate analysis. The corresponding hospitalization probability from initiation of third- or fourth-line was 30% with regorafenib versus 18% with TAS-102 at five weeks and 41% versus 28% at ten weeks, respectively. Hospitalizations irrespective of cause during regorafenib or TAS-102 therapy did neither impact median survival in patients undergoing only third-line therapy (never-hospitalized: 5.7 months [95% CI: 3.9–10.5] versus hospitalized: 5.4 months [95% CI: 2.8–9.6], *p* = 0.45), nor in patients receiving third- and fourth-line therapy (12.2 months [95% CI: 10.6–28.8] versus 18.6 months [95% CI: 6.3-not reached], *p* = 0.90). In conclusion, apart from poor ECOG performance status, regorafenib therapy was associated with an increased hospitalization probability during palliative systemic third- and fourth-line therapy in mCRC. However, hospitalizations during regorafenib or TAS-102 therapy did not impact OS beyond second-line therapy.

## 1. Introduction

Colorectal cancer is the second leading cause of cancer-related death in Europe [1] and worldwide [2]. By considering sidedness and molecular pathology in terms of a personalized treatment approach, a median overall survival (OS) of 28.0–38.3 and 18.3–23.0 months can be achieved in left-sided and right-sided metastatic colorectal cancer (mCRC), respectively [3].

After progression on fluorouracil, oxaliplatin, irinotecan, anti-VEGF and/or anti-EGFR (in case of RAS wild-type status) therapy, the oral tyrosine-kinase inhibitor regorafenib [4], as well as the cytotoxic drug combination trifluridine/tipiracil (TAS-102) [5], represent treatment options with a proven OS benefit against placebo. Regorafenib improved median OS from 5.0 months to 6.4 months (HR: 0.77) in the CORRECT trial [4]. TAS-102 increased median OS from 5.3 months to 7.1 months (HR: 0.68) in the RECOURSE trial [5]. Both drugs are approved by the Food and Drug Administration (FDA) and European Medicines Agency (EMA) and are recommended for the treatment of mCRC beyond second-line by the National Comprehensive Cancer Network (NCCN) [6] and European Society of Medical Oncology (ESMO) [7] guidelines. However, when evaluating the overall clinical benefit by the American Society of Clinical Oncology Value Framework (ASCO-VF) and ESMO Magnitude of Clinical Benefit Scale (MCBS), TAS-102 scores higher than regorafenib [8]. With incremental cost-effectiveness ratios of 395,223 USD and 399,740 USD per quality-adjusted life year, neither TAS-102 nor regorafenib are considered cost-effective from a United States payer’s perspective [9]. In terms of a “continuum of care” concept [10], regorafenib or TAS-102 are typically applied as third-line therapy and sequencing (regorafenib followed by TAS-102 or vice versa) may further increase OS improvements in mCRC. The different mode of action of regorafenib and TAS-102 is associated with a different toxicity profile. While fatigue and hand-foot-skin reactions are frequently reported with the use of regorafenib [4], hematotoxicity (especially neutropenia) is the major side effect of TAS-102 [5]. The highly differential toxicity profile of these substances has led to a potentially biased perception of drug tolerability and complications—such as hospitalization—in the oncologic community. Unplanned hospitalizations of patients with mCRC in the later course of the disease are frequent, necessitating valuable health system resources and causing additional financial toxicity besides drug expenses per se. Due to the lack of a direct comparison between regorafenib and TAS-102 in a randomized fashion, we intended to objectify possible differences in drug tolerability, using hospitalizations as clear-cut end point during treatment with regorafenib and TAS-102.

In this bicentric retrospective analysis, we aimed at investigating the frequency, duration, causes and hospitalization probability during regorafenib or TAS-102 therapy, as well as the impact of hospitalizations on clinical outcome in mCRC beyond second-line therapy.

## 2. Results

### 2.1. Baseline Characteristics

Between January 2013 and May 2019, 93 mCRC patients started systemic third-line therapy with regorafenib or TAS-102. The baseline characteristics are depicted in Table 1.

Sixty-nine patients and 24 patients received regorafenib and TAS-102 as third-line therapy, respectively, and 38 patients (41%) received regorafenib (*n* = 7) or TAS-102 (*n* = 31) as fourth-line therapy. In total, 76 patients were treated with regorafenib and 55 patients with TAS-102 during third- or fourth-line (Table 2).

The median time on therapy with regorafenib or TAS-102 during third-line and fourth-line was 64 days (range: 1–402 days) and 82 days (range: 10–413 days), respectively. The median time on therapy with regorafenib or TAS-102 did not differ during third-line (59 days (range: 1–402) versus 83 days (range: 4–366), *p* = 0.27) or fourth-line (median: 65 days (range: 15–159) versus 82 days (range: 10–413), *p* = 0.53). The median time from mCRC diagnosis to initiation of third-line therapy with either regorafenib or TAS-102, median time to follow-up from third-line start (for OS analysis), and median OS from start of third-line therapy were 21.5 months (range: 4.5–90.1), 20.5 months (95% CI: 12.7–28.3) and 10.4 months (95% CI: 6.7–12.2, Appendix A). OS from third-line therapy was not statistically significantly dependent on the administration of a subsequent therapy line, nor on the therapy sequence of regorafenib and TAS-102 (Appendix A). None of the patients with an Eastern Cooperative Oncology Group (ECOG) performance score ≥ 2 at the initiation of third-line therapy received a fourth-line therapy with regorafenib or TAS-102, whereas 54% of ECOG 0-1 patients were able to receive fourth-line therapy (0% versus 54%, *p* < 0.001).

### 2.2. Probability of Hospitalization during Third- and Fourth-Line Therapy

Among 93 mCRC patients undergoing third- or fourth-line therapy with regorafenib or TAS-102, 77 hospitalizations were necessitated in total. Forty-six percent (43 out of 93) and 34% (13 out of 38) of patients were hospitalized once or more during regorafenib or TAS-102 therapy during third- and fourth-line, respectively (Table 2). For covariate selection, a backward stepwise regression for competing risks regression was performed using the Akaike information criterion (AIC), as selection criterion including the following covariates: age at third-line therapy initiation, ECOG performance status at third-line initiation, sex, therapy line, sidedness, primary tumor resection status, time point of metastases detection (synchronous versus metachronous), RAS status, evidence of peritoneal metastases, liver metastases, and/or lung metastases at the start of third-line treatment, and tumor therapy (regorafenib versus TAS-102). The ECOG performance status, tumor therapy and sidedness were the selected covariates for the final regression model. The ECOG performance status (2–3 versus 0–1, HR 4.04 [95% CI: 2.11–7.71], *p* < 0.001) and tumor therapy (regorafenib versus TAS-102, HR: 1.95 [95% CI: 1.07–3.54], *p* = 0.03) showed a statistically significant association with the risk of hospitalization in multivariate analysis (Table 3).

The corresponding hospitalization probability from the initiation of third- or fourth-line was 30% with regorafenib versus 18% with TAS-102 at five weeks and 41% versus 28% at ten weeks, respectively (Figure 1). In a logistic regression analysis, no statistically significant association between hospitalization during third-line treatment and the probability to receive a fourth-line therapy was seen (*p* = 0.99).

### 2.3. Patient and Disease Characteristics and Risk of Hospitalization

Baseline characteristics at the initiation of third-line therapy did not statistically significantly differ between hospitalized and never-hospitalized mCRC patients undergoing only third-line therapy (*n* = 55) or third- and fourth-line therapy (*n* = 38), with regorafenib and TAS-102 or vice versa (Appendix A). CT images for the calculation of skeletal muscle mass/area and in turn for the detection of sarcopenia were available in 43 patients. Based on these CT images at the initiation of third-line therapy, sarcopenia was not associated with hospitalization probability in patients undergoing third-line therapy with regorafenib or TAS-102 without crossover in a subsequent therapy line (never-hospitalized: 64% sarcopenia versus hospitalized: 50% sarcopenia, *p* = 0.51). However, sarcopenic patients at third-line start tended to show a higher hospitalization risk among patients undergoing third- and fourth-line therapy with either regorafenib followed by TAS-102, or vice versa (never-hospitalized: 27% sarcopenia versus hospitalized: 67%, *p* = 0.08).

### 2.4. Length of Hospital Stay during Third- and Fourth-Line Therapy

The cumulative hospitalization nights during third- and fourth-line therapy (mean: regorafenib: 6.2 versus TAS-102: 5.3 days, *p* = 0.20), during third-line therapy (mean: regorafenib: 5.9 versus TAS-102: 8.0 days, *p* = 0.94) and fourth-line therapy (mean: regorafenib: 8.7 versus TAS-102: 3.2 days, *p* = 0.21) did not significantly differ between regorafenib and TAS-102 therapy.

### 2.5. Causes of Hospitalization

The causes of hospitalization during regorafenib or TAS-102 therapy are summarized according to therapy line in Table 4:

Gastrointestinal adverse events resulting in hospitalization were only observed with regorafenib during third- and fourth-line therapy (regorafenib: 15% versus TAS-102: 0%, *p* = 0.03). Other hospitalization causes were equally distributed between regorafenib and TAS-102 across therapy lines (Table 4).

### 2.6. Frequency of Hospitalization-Associated Therapy Discontinuation

Across third- and fourth-line therapy, hospitalized patients showed therapy discontinuation rates of 30% and 26% (at any hospitalization event) during regorafenib and TAS-102 therapy, respectively (*p* = 0.79). The frequency of hospitalization-associated therapy discontinuation did neither differ between regorafenib and TAS-102 during third-line (30% versus 20%, *p* = 0.53), nor fourth-line (25% versus 33%, *p* = 0.76) therapy (Table 5).

### 2.7. Impact of Hospitalization on Overall Survival

Hospitalizations did not impact OS in patients undergoing only third-line therapy (*n* = 55) with regorafenib or TAS-102 (median: never-hospitalized: 5.7 months [95% CI: 3.9–10.5] versus hospitalized: 5.4 months [95% CI: 2.8–9.6], *p* = 0.45, Figure 2A). No OS difference was found between never-hospitalized and hospitalized patients receiving third- and fourth-line therapy (*n* = 38) with regorafenib and TAS-102 or vice versa (median: 12.2 months [95% CI: 10.6–28.8] versus 18.6 months [95% CI: 6.3-not reached], *p* = 0.90, Figure 2B).

## 3. Discussion

To the best of our knowledge, this is the first report on hospitalizations and hospitalization-related clinical outcomes during palliative systemic therapy with regorafenib and/or TAS-102 in mCRC in a real-world setting. Apart from the ECOG performance status at third-line initiation, tumor therapy had a statistically significantly independent impact on hospitalization probability during third- and fourth-line therapy (Table 3). In line with the toxicity profile, hospitalizations due to gastrointestinal toxicity during third- and fourth-line therapy were only observed with regorafenib (Table 4). The hospitalization rate due to infections did not differ between TAS-102 and regorafenib (Table 4). However, the rate of febrile neutropenia was not assessed in this retrospective analysis. Despite hospitalization-associated drug discontinuation rates ranging from 26% to 30% (Table 5), hospitalizations during regorafenib or TAS-102 therapy did not impact OS (Figure 2).

With a median OS of 9.6 and 10.4 months in patients only receiving third-line therapy with either regorafenib or TAS-102 (Appendix A), our bicentric real-world mCRC cohort showed a superior clinical outcome in comparison to the landmark trials CORRECT [4] and RECOURSE [5]. The ECOG performance status at the initiation of third-line therapy had a considerable impact on treatment beyond second-line, as none of the ECOG 2 and 3 patients received sequential therapy with regorafenib and TAS-102 or vice versa. It is noteworthy that inclusion in the CORRECT [4] and RECOURSE [5] trial was restricted to patients with an ECOG performance status ≤ 1.

In consideration of 77 hospital admissions among 93 patients, hospitalizations during regorafenib and TAS-102 therapy were frequent events in our mCRC cohort. Kotani et al. reported an emergency hospitalization rate of 24% during TAS-102 therapy, however, more than three quarters of the events were disease-related [11]. According to a retrospective analysis by Calcagno et al. including 29 mCRC patients, hospitalizations during regorafenib treatment were necessitated in only 10%, which were all attributable to drug-related adverse events (rash, bleeding, heart failure) [12]. While regorafenib-specific hand-foot-skin reactions were manageable in an outpatient setting, seven admissions were caused by regorafenib-induced gastrointestinal toxicity in our mCRC cohort (Table 4). Comparing baseline characteristics at the time point of third-line initiation between hospitalized and never-hospitalized patients could not identify patient subgroups at risk for hospitalization (Appendix A). We previously published that more than half of mCRC patients present with decreased skeletal muscle mass, so-called “sarcopenia”, at the start of third-line therapy [13]. Sarcopenia increases the likelihood of hospitalization among older people [14]. Cross-sectional CT-images for assessment of sarcopenia were available in 43 patients. Sarcopenia at the initiation of third-line therapy was not associated with the risk of hospitalization in mCRC patients receiving only third-line therapy with either regorafenib or TAS-102 (*p* = 0.51, Appendix A). There was only a trend towards sarcopenia-associated hospitalization in mCRC patients undergoing sequential therapy with regorafenib and TAS-102 (or vice versa) (*p* = 0.08, Appendix A). However, due to the limited number of patients with available CT images, the interpretation of these findings has to be done with caution.

With a median time of 2.1 months and 2.7 months during third- and fourth-line therapy in our mCRC cohort, time on treatment was as short-lived as in the CORRECT [4] and RECOURSE [5] trial. Despite the short duration on therapy, the hospitalization probability ranged from 18–30% at five weeks and 28–41% at ten weeks, respectively (Figure 1). Due to the fact that the main cause for hospitalization was disease-related (Table 4), electronic patient-reported outcome (PRO) monitoring with alerts to clinicians or nurses may serve as a tool to avoid or delay hospitalizations in mCRC outpatients on regorafenib or TAS-102. In a randomized controlled trial, Basch et al. demonstrated the feasibility of symptom monitoring via tablet computers in patients receiving outpatient palliative systemic therapy for advanced solid tumors. Pre-specified symptom worsening, as well as symptom severity surpassing a pre-specified threshold, triggered an e-mail alert to nurses. Symptom management and supportive medication initiation/modification statistically significantly improved health-related quality of life (QoL) and resulted in fewer emergency room visits in the intervention group [15]. In an updated analysis, Basch et al. reported an OS benefit (secondary endpoint) for patients randomized to electronic PRO monitoring [16]. Patients undergoing regorafenib or TAS-102 treatment should be at least scheduled for biweekly visits during therapy cycles 1 and 2 and every four weeks during the following cycles [4,5]. With regard to the toxicity profile of regorafenib, electronic PRO monitoring could serve as tool to counteract common adverse events, including diarrhea, hand-foot-skin reaction and loss of appetite and in turn delay or avoid hospitalizations. The predominant cause of hospitalization in our mCRC patient cohort undergoing regorafenib or TAS-102 treatment was related to the underlying malignancy—in this regard—electronic PRO monitoring could offer the opportunity to intervene (e.g., by dose modifications of analgetics or antiemetics) in both oral therapy strategies.

Our analysis has several limitations: The reported findings such as e.g., individual hospitalization causes might have been biased by the retrospective nature of the study. Twenty patients (22%) received subsequent systemic therapy after regorafenib and/or TAS-102, which may have impacted on OS results. Furthermore, QoL data and the impact of hospitalizations on QoL were not assessed, and the ECOG performance status was not available for each patient, due to the retrospective character of the study. Lastly, the number of included patients (*n* = 93) was limited, despite a bi-centric approach.

## 4. Materials and Methods

This retrospective analysis was based on unselected consecutive mCRC patients treated with regorafenib and/or TAS-102 at the tertiary cancer centers in Salzburg and Wels-Grieskirchen (Austria). Analyses were approved by the Ethics Committee of the provincial government of Salzburg, Austria (415-EP/73/655–2016). All included patients alive at the date of analysis signed an informed consent form. Prior disease progression on fluorouracil, oxaliplatin, irinotecan, anti-VEGF and/or anti-EGFR (in case of RAS wild-type status) targeted therapy was a prerequisite for the initiation of regorafenib and/or TAS-102 therapy. Early access to regorafenib and/or TAS-102 within a named patient program was available for patients who had received regorafenib and/or TAS-102 before the respective approval by the EMA. TAS-102 was orally applied twice daily at a dose of 35 mg/m^2^ five days a week, with 2 days of rest, for 2 weeks, followed by a 14-day rest period, and repeated every four weeks [5]. Regorafenib was either prescribed at an oral daily dose of 160 mg for the first three weeks of each four-week cycle [4], or at a starting dose of 80 mg per day with weekly dose escalation to a target dose of 160 mg [17].

Baseline patient characteristics at the start of third-line therapy with either regorafenib or TAS-102 were retrospectively assessed. Data were extracted from medical records, including hospitalization frequency, hospitalization duration, cause of hospitalization, drug discontinuation rate due to hospitalization, age, sex, Eastern Cooperative Oncology Group (ECOG) performance status, sarcopenia assessed by cross-sectional CT images at the level of the third lumbar vertebra using the manual method [13], using sex-specific cut-off values [18] (men: skeletal muscle index < 52.4 cm^2^/m^2^; women: skeletal muscle index < 38.5 cm^2^/m^2^), third- and fourth-line cancer therapy (regorafenib, TAS-102), subsequent systemic therapy, time point of detection of metastases (synchronous versus metachronous), primary tumor resection status, primary tumor location, metastatic pattern, presence of ascites, RAS status, BRAF status and microsatellite status. Differences in patient baseline characteristics between two groups were tested by Pearson’s χ2-test. For continuous data, the difference between the two groups was calculated with the two-sided Wilcoxon rank-sum test.

The primary outcome of our analysis was cumulative risk for hospitalization during treatment with regorafenib and TAS-102 as third- or fourth-line therapy, respectively. The first hospitalization with a hospital admission for at least one night was the event of interest. If a patient was treated sequentially with both drugs (regorafenib followed by TAS-102 or TAS-102 followed by regorafenib), the patient was considered twice for this analysis, for third- and fourth line treatment with the respective drug. A Fine–Gray regression model was used for competing risk analysis. Multivariate analysis was based on a Fine–Gray proportional subdistribution hazards regression model. For multivariate analysis covariate selection, a backward stepwise regression for competing risks regression was performed using AIC as selection criterion. OS was calculated from the start of third-line treatment until the date of death or date of last known follow-up. Patients alive at the last contact were censored. Survival curves were estimated by the Kaplan–Meier method. For survival analysis, median follow-up was calculated from initiation of third-line treatment with either regorafenib or TAS-102 using the Kaplan–Meier estimator with reversed status indicators (death and censored). For survival analysis according to treatment groups, adjusted survival curves using Cox proportional hazards models were created. The likelihood-ratio test was used to compare survival distributions between patient groups. In order to avoid immortal time bias, cross-over was taken into account as time-dependent covariate beginning with the start of fourth-line treatment.

All statistical analyses were performed using IBM SPSS Version 23 and the statistical software environment R (packages cmprsk, survival).

## 5. Conclusions

In summary, this is the first report on the hospitalization probability and hospitalization-associated clinical outcome in mCRC patients undergoing treatment with regorafenib or TAS-102 beyond second-line. We found a higher hospitalization probability beyond-second line during regorafenib treatment, as well as in patients with a poor ECOG performance status. However, hospitalizations irrespective of cause during third- or fourth-line therapy with regorafenib or TAS-102 had no impact on clinical outcome. Electronic PRO monitoring may serve as a tool for the early detection of cancer-related symptom worsening, and to intervene in order to delay or avoid hospitalizations during regorafenib or TAS-102 treatment. In light of increased gastrointestinal toxicity leading to hospitalization during regorafenib treatment, we call for increased awareness to drug-specific toxicities, in order to prevent unnecessary complications by early detection of adverse events and prompt counteraction.

## Figures and Tables

**Figure 1 cancers-12-02812-f001:**
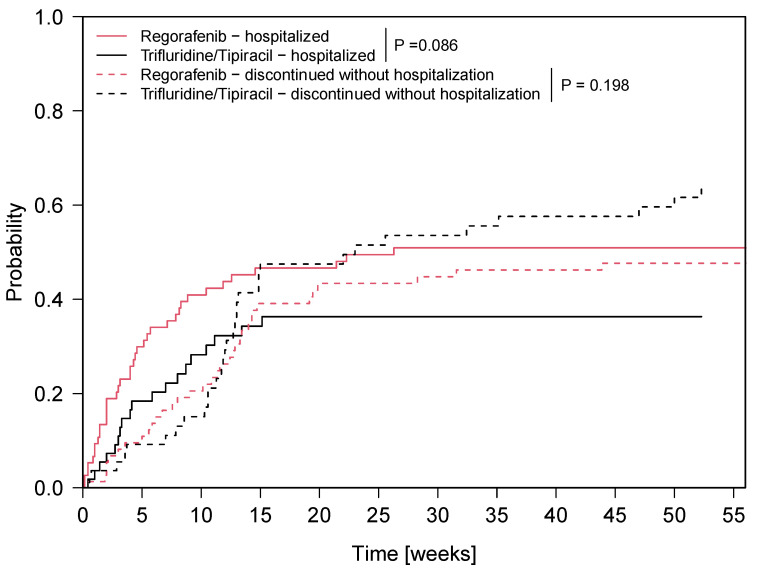
Cumulative incidence (event: first hospitalization) analysis with regorafenib and TAS-102 during third- and fourth-line therapy. y-axis: hospitalization probability, x-axis: time (weeks) from initiation of third-line or fourth-line therapy with regorafenib or TAS-102.

**Figure 2 cancers-12-02812-f002:**
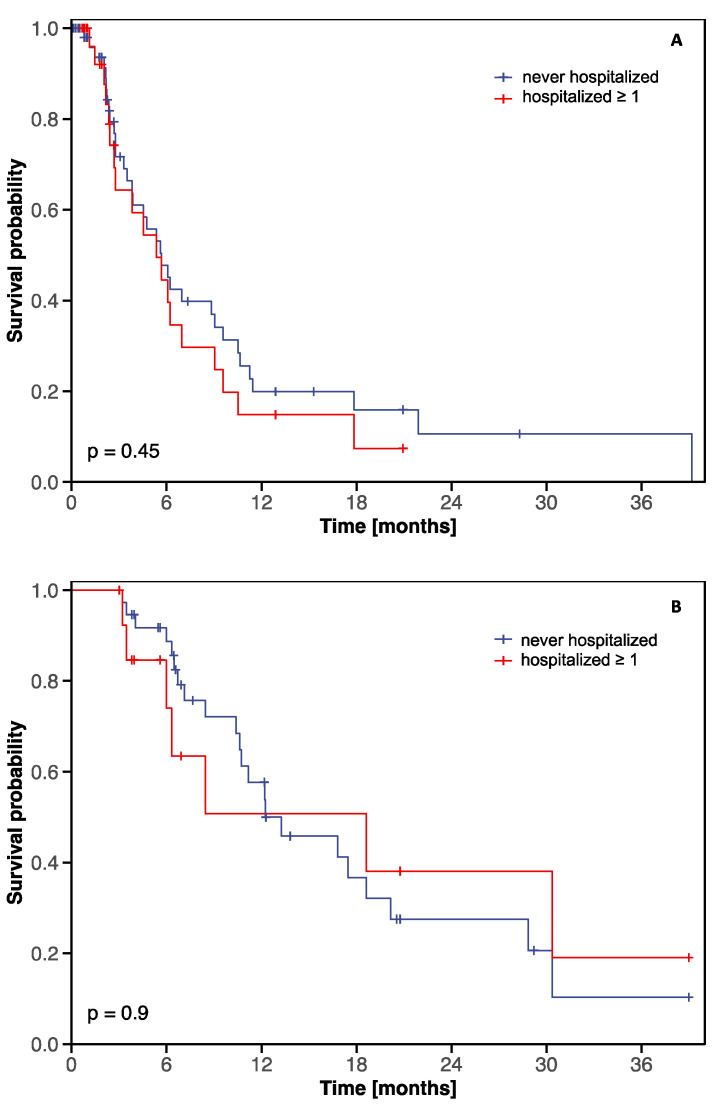
Overall survival from initiation of third-line therapy according to hospitalization status (hospitalized ≥ 1 versus never hospitalized) during regorafenib or TAS-102 therapy. (**A**) OS in 55 mCRC patients undergoing third-line therapy with regorafenib or TAS-102 without subsequent cross-over. (**B**) OS in 38 mCRC patients undergoing third- and fourth line therapy with regorafenib and TAS-102 (or vice versa). The tick marks on the curves represent censored patients. Hospitalization was taken into account as a time-dependent covariate, in order to avoid immortal time bias.

**Table 1 cancers-12-02812-t001:** Baseline characteristics of 93 metastatic colorectal cancer (mCRC) patients receiving regorafenib and/or TAS-102 during third- or fourth-line therapy.

Parameter		All *n* = 93 (%)	Regorafenib 3rd Line *n* = 69 (%)	TAS-102 3rd Line *n* = 24 (%)	*p*-Value
Age at 3rd line start	median (range)	65 (42–85)	65 (42–85)	68 (49–81)	0.11 *
Tertiary cancer center	Salzburg Wels-Grieskirchen	60 (65) 33 (35)	43 (62) 26 (38)	17 (71) 7 (29)	0.45
Sex	male female	54 (58) 39 (42)	40 (58) 29 (42)	14 (58) 10 (42)	0.98
ECOG PS at 3rd line start	0 1 2 3 NA	25 (35) 34 (47) 11 (15) 2 (3) 21	21 (40) 25 (47) 6 (11) 1 (2) 16	4 (21) 9 (48) 5 (26) 1 (5) 5	0.26
Detection of metastases	synchronous metachronous	62 (67) 31 (33)	49 (71) 20 (29)	13 (54) 11 (46)	0.13
Primary tumor resected	yes no	75 (81) 18 (19)	54 (78) 15 (22)	21 (88) 3 (12)	0.32
Sidedness	left right	67 (72) 26 (28)	47 (68) 22 (32)	20 (83) 4 (17)	0.15
Metastatic pattern at 3rd line start	liver^+^ lung^+^ peritoneum^+^	72 (77) 64 (69) 16 (17)	56 (81) 49 (71) 10 (14)	16 (67) 15 (63) 6 (25)	0.14 0.44 0.24
Ascites at 3rd line start	yes no	8 (9) 85 (91)	4 (6) 65 (94)	4 (17) 20 (83)	0.10
RAS status	wild-type mutant	46 (49) 47 (51)	34 (49) 35 (51)	12 (50) 12 (50)	0.95
BRAF status	wild-type mutant NA	68 (99) 1 (1) 24	50 (100) 0 (0) 19	18 (95) 1 (5) 5	0.10
Microsatellite status	MSS MSI NA	58 (97) 2 (3) 33	44 (98) 1 (2) 24	14 (93) 1 (7) 9	0.41
Subsequent therapy with regorafenib or TAS-102	Regorafenib TAS-102		- 31 (45)	7 (29) -	0.18
Subsequent other systemic therapy after regorafenib and/or TAS-102	yes no	20 (22) 73 (78)	16 (23) 53 (77)	4 (17) 20 (83)	0.50

mCRC: metastatic colorectal cancer, ECOG PS: Eastern Cooperative Oncology Group performance status, MSI: microsatellite instability, MSS: microsatellite stability, NA: not available, * Wilcoxon rank-sum test, ^+^multiple designations are possible.

**Table 2 cancers-12-02812-t002:** Hospitalized patients and total hospitalizations during regorafenib and TAS-102 sequencing in third- and fourth-line therapy in 93 mCRC patients.

3rd Line (93 Patients)	REGO *n* = 69 (%)		TAS-102 *n* = 24 (%)	
Number of hospitalizations (*n* = 59)	42		17	
Number of hospitalized patients (*n* = 43, 46%)	33 (48)		10 (42)	
Range of hospitalizations per patient	0–3		0–4	
**4th Line (38 Patients)**	**TAS-102** ***n* = 31 (%)**	**No TAS-102** ***n* = 38**	**REGO** ***n* = 7 (%)**	**No REGO** ***n* = 17**
Number of hospitalizations (*n* = 18)	13	-	5	-
Number of hospitalized patients (*n* = 13, 34%)	9 (29)	-	4 (57)	-
Range of hospitalizations per patient	0–4	-	0–2	-

REGO: regorafenib.

**Table 3 cancers-12-02812-t003:** Multivariate analysis (risk of hospitalization) during third- and fourth-line therapy.

Covariates	HR (95% CI)	*p*-Value
tumor therapy (regorafenib versus TAS-102)	1.95 (1.07–3.54)	0.03
ECOG performance status at third-line start (2–3 versus 0–1)	4.04 (2.11–7.71)	< 0.001
sidedness (right versus left)	1.64 (0.91–2.95)	0.1

HR: hazard ratio, ECOG: Eastern Cooperative Oncology Group.

**Table 4 cancers-12-02812-t004:** Hospitalization causes during third- and fourth-line therapy with regorafenib and TAS-102.

Cause	3rd + 4th Line Number of Hospitalizations (*n* = 77)			3rd Line Number of Hospitalizations (*n* = 59)			4th Line Number of Hospitalizations (*n* = 18)		
	REGO *n* = 47 (%)	TAS *n* = 30 (%)	*p*-value	REGO *n* = 42 (%)	TAS *n* = 17 (%)	*p*-value	REGO *n* = 5 (%)	TAS *n* = 13 (%)	*p*-value
Infection	14 (30)	8 (27)	0.77	12 (29)	4 (23)	0.69	2 (40)	4 (31)	0.71
Disease-related	24 (51)	20 (67)	0.18	22 (52)	12 (71)	0.20	2 (40)	8 (61)	0.41
GI-toxicity	7 (15)	0 (0)	**0.03***	6 (14)	0 (0)	0.10	1 (20)	0 (0)	0.10
Other	2 (4)	2 (6)	0.64	2 (5)	1 (6)	0.86	0 (0)	1 (8)	0.52

GI: gastrointestinal, REGO: regorafenib, TAS: TAS-102. *indicating statistical significance.

**Table 5 cancers-12-02812-t005:** Hospitalization-associated regorafenib or TAS-102 discontinuation.

Therapy Discontinuation Due to Any Hospitalization	Regorafenib	TAS-102	*p*-Value
Number of hospitalized patients during 3rd + 4th line: *n* = 56			
no yes	26 (70) 11 (30)	14 (74) 5 (26)	0.79
Number of hospitalized patients during 3rd line: *n* = 43			
no yes	23 (70) 10 (30)	8 (80) 2 (20)	0.53
Number of hospitalized patients during 4th line: *n* = 13			
no yes	3 (75) 1 (25)	6 (67) 3 (33)	0.76

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
