# Peer review of "Hospitalizations and Clinical Outcome in Metastatic Colorectal Cancer During Regorafenib or TAS-102 Therapy"

_cancers, 2020, doi:10.3390/cancers12102812_

Round 1

Reviewer 1 Report

The manuscript by Huemer and coworkers is well-written and the methodology is well presented. The authors provide a new clinical inside about metastatic colorectal cancer (mCRC) patients treated beyond second line of treatment. They analyze hospitalization rate in patients receiving regorafenib or TAS-102 in third- or fourth-line setting, as recommended per clinical guidelines. They found that there was a trend toward a higher hospitalization rate in patients receiving regorafenib as a third-line therapy but statistically significance was not reached. Furthermore, they reported no impact of hospitalizations in overall survival (OS). The authors interestingly provide a new aspect to be considered in terms of cost-effectiveness of an advance line of treatment in mCRC. However, some concerns on the submitted manuscript emerged.

  • The real impact of this manuscript on clinical decision making in later lines of treatment for mCRC has to be defined. Indeed, the authors did not report a statistically relevant increase of hospitalization in patients receiving regorafenib as a third-line treatment. Furthermore, hospitalization did not impact on OS. However, the authors suggest the use of electronic patient-reported outcome (PRO). How can the authors speculate that PRO might be helpful in reducing hospitalization? The discussion section and in particular the paragraph on PRO is too long and has to be shortened.
  • The different rate of hospitalization between regorafenib and TAS-102 might be altered by the fact that while 69 patients received regorafenib as a third-line therapy, only 24 were treated with TAS-102. Authors should mention this aspect in the discussion. I suggest adding to the discussion a paragraph to discuss the limitations of this manuscript such as retrospective nature of the research, lack of quality of life assessment and discrepancy between the number of patients who received regorafenib and TAS-102.
  • How many patients received a further line of treatment beyond regorafenib or TAS-102 (i.e. retreatment with anti-EGRF antibodies or other cytotoxic agents?). This might impact on OS.
  • The author state that hospitalization did not impact on OS. Did hospitalization during third-line treatment impact on the probability to receive a fourth-line treatment?
  • In the multivariate regression model the authors did not include peritoneal involvement which is well-known poor prognostic factor affecting 17 patients of the present cohort (reported in Table 1). Might it be involved in increasing the chance of hospitalization? I would recommend including peritoneal involvement in multivariate regression model especially considering that most of the cause of hospitalization were disease-related (Figure 2).

Author Response

We thank reviewer 1 for the valuable input, which has definitely helped to improve our manuscript. The raised questions are answered in a point-by-point fashion and changes in the revised manuscript are highlighted via “track-change mode”.

Rievewer #1

„The manuscript by Huemer and coworkers is well-written and the methodology is well presented. The authors provide a new clinical inside about metastatic colorectal cancer (mCRC) patients treated beyond second line of treatment. They analyze hospitalization rate in patients receiving regorafenib or TAS-102 in third- or fourth-line setting, as recommended per clinical guidelines. They found that there was a trend toward a higher hospitalization rate in patients receiving regorafenib as a third-line therapy but statistically significance was not reached. Furthermore, they reported no impact of hospitalizations in overall survival (OS). The authors interestingly provide a new aspect to be considered in terms of cost-effectiveness of an advance line of treatment in mCRC. However, some concerns on the submitted manuscript emerged.“

Reviewer #1 point #1:

“The real impact of this manuscript on clinical decision making in later lines of treatment for mCRC has to be defined. Indeed, the authors did not report a statistically relevant increase of hospitalization in patients receiving regorafenib as a third-line treatment. Furthermore, hospitalization did not impact on OS. However, the authors suggest the use of electronic patient-reported outcome (PRO). How can the authors speculate that PRO might be helpful in reducing hospitalization? The discussion section and in particular the paragraph on PRO is too long and has to be shortened.”

Authors’ response:

According to the reviewer’s suggestion we elucidated the rationale for this analysis (highly differential toxicity profile of these substances and potentially biased perception of drug tolerability and complications - such as hospitalization - in the oncologic community) in the “Introduction” section. Furthermore, we elucidated the clinical impact of our findings in the “Conclusions” section (In light of increased gastrointestinal toxicity leading to hospitalization during regorafenib treatment, we call for increased awareness to drug-specific toxicities in order to prevent unnecessary complications by early detection of adverse events and prompt counteraction).

We acknowledge the reviewer’s comment concerning the application of PRO monitoring. The application of PRO monitoring during systemic therapy in advanced solid tumors has already proven effective in symptom-management and supportive medication initiation/modification in a randomized controlled trial. The latter study demonstrated significant improvements of health-related quality of life and fewer emergency room visits in the PRO monitoring intervention arm (Basch et al., JCO 2016) and even showed an OS benefit (secondary endpoint, Basch et al. JAMA 2017). Due to the fact that the most frequent cause of hospitalization was disease-related in our mCRC cohort beyond second-line (Table 4), we hypothesize that PRO monitoring might serve as a tool to delay or even prevent hospitalizations by e.g. adjustment of pain or antiemetic mediation. However, our results do not provide evidence for PRO measures - therefore – the statement concerning PRO monitoring has been formulated more carefully in the “Discussion” section. As suggested, the „PRO monitoring“ part in the “Discussion” section has been considerably shortened.

Reviewer #1 point #2:

„The different rate of hospitalization between regorafenib and TAS-102 might be altered by the fact that while 69 patients received regorafenib as a third-line therapy, only 24 were treated with TAS-102. Authors should mention this aspect in the discussion. I suggest adding to the discussion a paragraph to discuss the limitations of this manuscript such as retrospective nature of the research, lack of quality of life assessment and discrepancy between the number of patients who received regorafenib and TAS-102.“

Authors’ response:

As suggested by the reviewer, a separate paragraph has been added in the “Discussion” section addressing limitations such as the retrospective nature of the analysis, lack of quality of life (QoL) data as well as the impact of hospitalizations on QoL and the limited patient number included. We fully agree with the reviewer about the imbalance of absolute numbers of patients starting third-line and and/or fourth line therapy with either regorafenib or TAS-102. However, assessment of hospitalization probability (percentage) takes the number of patients on the respective drug into account, therefore, the hospitalization probability should not be affected by the absolute number of patients treated with regorafenib or TAS-102.  We would like to emphasize that another reviewer suggested to show and compare the baseline characteristics of our mCRC patients stratified according to third-line therapy (regorafenib versus TAS-102). The latter information has been integrated into “Table 1” without providing evidence for statistically significant imbalances between the two groups.

Reviewer #1 point #3:

„How many patients received a further line of treatment beyond regorafenib or TAS-102 (i.e. retreatment with anti-EGRF antibodies or other cytotoxic agents?). This might impact on OS.“

Authors’ response:

We appreciate the reviewer’s comment on a putative impact of subsequent therapy lines on overall survival. The subsequent systemic treatment status following regorafenib or TAS-102 has been added to “Table 1” (which has been stratified according to regorafenib and TAS-102 therapy in the meantime as requested by another reviewer). In total, 20 out of 93 (22%) mCRC patients received subsequent palliative systemic therapy after regorafenib and/or TAS-102 treatment. We did not detect any imbalances concerning subsequent systemic therapy after regorafenib and/or TAS-102 treatment between mCRC patients starting third-line therapy with either regorafenib or TAS-102 (23% versus 17%, p=0.50) (Table 1). We cannot exclude a bias on OS from subsequent therapies, however, this issue has been addressed in the limitations paragraph of the “Discussion” section.

Reviewer #1 point #4:

“The author state that hospitalization did not impact on OS. Did hospitalization during third-line treatment impact on the probability to receive a fourth-line treatment?”

Authors’ response:

According to the reviewer’s suggestion we tested whether hospitalization during third-line therapy with regorafenib or TAS-102 influenced the probability to receive a fourth-line therapy. In a logistic regression analysis (using R package glm), no statistically significant association between hospitalization during third-line treatment and the application of a fourth-line therapy was seen (p=0.99). The latter finding has been added to the “Results” section (2.2. Probability of hospitalizations during third- and fourth-line therapy).

Reviewer #1 point #5:

“In the multivariate regression model the authors did not include peritoneal involvement which is well-known poor prognostic factor affecting 17 patients of the present cohort (reported in Table 1). Might it be involved in increasing the chance of hospitalization? I would recommend including peritoneal involvement in multivariate regression model especially considering that most of the cause of hospitalization were disease-related (Figure 2).“

Authors’ response:

We appreciate the reviewer’s suggestion to test whether evidence of peritoneal metastases impacts hospitalization probability.

First, the impact of evidence of peritoneal metastases on hospitalization probability has been tested in a stepwise backward regression model with other covariates (age at third-line initiation, ECOG performance status (0-1 versus 2-3) at third-line initiation, sex, therapy line, sidedness, primary tumor resection status, time point of metastases detection (synchronous versus metachronous), evidence of pulmonary metastases, evidence of liver metastases, RAS status, and tumor therapy (regorafenib versus TAS-102)). The ECOG performance status, tumor therapy and sidedness were selected covariates for the final regression model. The ECOG performance status (2-3 versus 0-1, HR: 4.04 [95% CI: 2.11-7.71], p<0.001) as well as tumor therapy (regorafenib versus TAS-102, HR: 1.95 [95% CI: 1.07-3.54], p=0.03) remained statistically significantly associated with hospitalization probability, but this was not the case for sidedness (right versus left, HR: 1.64 [95%CI: 0.91-2.95], p=0.10). These findings including the below shown MVA analysis table (Table 3 in the revised manuscript) have been added to the „Results“ section (2.2. Probability of hospitalization during third- and fourth-line therapy)

Multivariate analysis using covariates selected by a stepwise backward regression model:

HR (95% CI)

p-value

tumor therapy (regorafenib versus TAS-102

1.95 (1.07, 3.54)

0.03

ECOG performance status at 3rd line start (2-3 versus 0-1)

4.04 (2.11, 7.71)

<0.001

sidedness (right versus left)

1.64 (0.91, 2.95)

0.1

In a second approach (data not included in the manuscript), evidence of peritoneal metastases has been tested in univariate analysis where no association with hospitalization probability was found (absence versus presence of peritoneal metastases at third-line start, HR: 0.77 [95% CI: 0.36-1.65], p=0.50). When evaluating the impact of evidence of peritoneal metastases on hospitalization probability in multivariate analysis with other covariates (irrespective of statistical significance), peritoneal metastases did not influence hospitalizations. However, we would like to emphasize that a stepwise backward regression model for covariate selection was used in order to prevent imbalances between the number of tested covariates and hospitalization events.

Multivariate analysis (hospitalization probability) irrespective of statistical significance (only for the reviewer, not included in the final manuscript):

HR (95% CI)

p-value

therapy line (third- versus fourth-line)

1.21 (0.49, 2.98)

0.69

age (continuous variable)

1.01 (0.97, 1.05)

0.75

sex (male versus female)

1.49 (0.69, 3.24)

0.31

RAS status (wild-type versus mutant)

0.56 (0.27, 1.14)

0.11

metachronous versus synchronous detection

0.62 (0.32, 1.21)

0.16

primary tumor resection status (yes versus no)

1.02 (0.38, 2.75)

0.96

sidedness (right versus left)

2.02 (0.9, 4.54)

0.09

tumor therapy (regorafenib versus TAS-102)

2.03 (0.98, 4.2)

0.057

Evidence of pulmonary metastases (no versus yes)

0.92 (0.49, 1.71)

0.79

Evidence of liver metastases (no versus yes)

0.89 (0.39, 2)

0.77

Evidence of peritoneal metastases (no versus yes)

0.6 (0.25, 1.49)

0.27

ECOG performance status (2-3 versus 0-1)

5.48 (2.41, 12.47)

<.001

Reviewer 2 Report

TAS-102 improves overall survival in patients with mCRC refractory to standard treatments. This study represents a new and interesting source of informations on the need for hospitalization of mCRC patients during the treatment and about the impact of hospitalization on clinical outcome.

Author Response

We thank reviewer 2 for reviewing our manuscript and the positive feedback. No suggestions for improvement have been made.

Rievewer #2

„TAS-102 improves overall survival in patients with mCRC refractory to standard treatments. This study represents a new and interesting source of informations on the need for hospitalization of mCRC patients during the treatment and about the impact of hospitalization on clinical outcome.“

Reviewer 3 Report

The current study aimed at evaluating and comparing hospitalization rates among metastatic colorectal cancer patients treated with either regorafenib or TAS-102 beyond second-line therapy. Moreover, the impact of hospitalizations of overall survival was also evaluated.

The study would be interesting from a clinical point of view, adding important information on the clinical impact of such therapies in the real-world clinical practice. However, I have several doubts and comments that I would like to be addressed. First of all, the following two important points, that would impact on the statistical analysis plan:

  1. In relation to the primary outcome (i.e. the occurrence of hospitalizations), it is not clear to me whether the outcome of interest is the first hospitalization occurred for each patients, or the total number of hospitalizations occurred during follow-up for each patient;
  2. It is unclear to me how you considered the exposure to regorafenib or TAS-102 during follow-up: did you employed an intention-to-treat design, by classifying patients as always exposed to the treatment received as third-line therapy, or you also considered switching from one therapy to another during follow-up? For example, in Figure 2, did you compare 69 and 24 patients treated, respectively, with third-line regorafenib or TAS-102, or you also included in the regorafenib group those patients who switched from TAS-102 to regorafenib?

Moreover, the following comments:

  1. Abstract and Results: if stated that “a trend towards a higher hospitalization probability with regorafenib compared to TAS-102 (HR 0.62 [95% CI: 0.36 – 1.06])”, the HR should be inverted (i.e. the reference group should be TAS-102, and the HR should be greater than 1)
  2. In Table1 would be interesting to report and compare baseline characteristics stratified by therapy (i.e. regorafenib of TAS-102)
  3. In Table2, please add percentage (%)
  4. Results: line 97: what do you mean with “median time of follow-up from third-line start”? Please define the follow-up period
  5. Figure S2: the number of events seems to be too small for any statistical inference on the comparison of survival between groups
  6. Figure S3: when comparing single-agent vs sequenced therapy, patients treated with sequenced therapy are affected by “immortal time” bias. Indeed, the time between initiation of third-line and initiation of fourth-line therapy is, by definition, “immortal”. This bias underestimates the survival of patients treated with sequenced therapy
  7. Similarly, Figure 3a/3b: in patients who experienced an hospitalization, the time from start of therapy to hospitalization is “immortal time” by definition, making the results biased
  8. Results: the backward stepwise regression selected only the treatment received (i.e. regorafenib or TAS-102) as covariate to be included in the final model. This may be due to the small number of patients included in the study cohort, which may be associated to a higher random variability (i.e. wide confidence intervals and low statistical power to detect significant associations). I found weird that age (for instance) is not associated to the risk of hospitalization. I would suggest to add to the model age and sex as covariates, even if not statistical significant
  9. Figure 1: as the main outcome of interest is the hospitalization occurrence, I would suggest to delete the discontinuation without hospitalization curves from the figure, and to truncate the follow-up at 30 weeks, since no events occurred after that and given (I guess) the low number of patients at risk after 30 weeks.
  10. What does Figure 2 add to the results reported in Table 3?

Author Response

We thank reviewer 3 for the valuable input, which has definitely helped to improve our manuscript. The raised questions are answered in a point-by-point fashion and changes in the revised manuscript are highlighted via “track-change mode”.

Rievewer #3

“The current study aimed at evaluating and comparing hospitalization rates among metastatic colorectal cancer patients treated with either regorafenib or TAS-102 beyond second-line therapy. Moreover, the impact of hospitalizations of overall survival was also evaluated.

The study would be interesting from a clinical point of view, adding important information on the clinical impact of such therapies in the real-world clinical practice. However, I have several doubts and comments that I would like to be addressed. First of all, the following two important points, that would impact on the statistical analysis plan:”

Reviewer #3 point #1:

“In relation to the primary outcome (i.e. the occurrence of hospitalizations), it is not clear to me whether the outcome of interest is the first hospitalization occurred for each patients, or the total number of hospitalizations occurred during follow-up for each patient;“

Authors’ response:

We appreciate the reviewer’s question on this point. In the meanwhile, we have specified the outcome parameters in Table 2 and present outcome data according to therapy line as well as according to systemic therapy: 1) the total number of hospitalization (a single patient could be hospitalized several times), 2) the number of hospitalized patients (at least one hospitalization necessitated), 3) the percentage of hospitalized patients (≥1) as well as 4) the range of the number of hospitalizations per patient. The primary outcome in Figure 1 (competing risk analysis) is hospitalization probability, representing patients’ first hospitalization, assessing whether an individual patient was hospitalized or never-hospitalized during regorafenib versus TAS-102 treatment. Table 4 (former Table 3) shows the total number of hospitalizations and the respective hospitalization causes during third- and fourth-line therapy with regorafenib and TAS-102. Please note that an individual patient could have been hospitalized several times during regorafenib or TAS-102 therapy and as a consequence could “contribute” with multiple hospitalizations to Table 4.  Data presented in Table 5 depict whether any hospitalization in hospitalized patients (an individual patient could have been hospitalized several times) caused therapy discontinuation during regorafenib or TAS-102 treatment.

We made efforts to specify the respective outcome in the figure and table legends as well as in the text.

Reviewer #3 point #2:

“It is unclear to me how you considered the exposure to regorafenib or TAS-102 during follow-up: did you employed an intention-to-treat design, by classifying patients as always exposed to the treatment received as third-line therapy, or you also considered switching from one therapy to another during follow-up? For example, in Figure 2, did you compare 69 and 24 patients treated, respectively, with third-line regorafenib or TAS-102, or you also included in the regorafenib group those patients who switched from TAS-102 to regorafenib?”

Authors’ response:

We appreciate the reviewer’s comment on this point. When comparing TAS-102 and regorafenib, switching from one therapy to another was always taken into account. In other words, in former Figure 2 (which has been removed according to the reviewer’s suggestions in the meanwhile due to redundancy with Table 4 (former Table 3)) all hospitalization causes during regorafenib (3rd line: n=69 patients + 4th line: n=7 patients) and TAS-102 (3rd line: n=24 patients + 4th line: n=31 patients) across 3rd and 4th line have been compared. Comparison of hospitalization causes during actual regorafenib and TAS-102 treatment period is shown according to separate treatment lines (3rd line as well as 4th line) as well as across 3rd line and 4th line in Table 4 (former Table 3).

Reviewer #3 point #3:

“Abstract and Results: if stated that “a trend towards a higher hospitalization probability with regorafenib compared to TAS-102 (HR 0.62 [95% CI: 0.36 – 1.06])”, the HR should be inverted (i.e. the reference group should be TAS-102, and the HR should be greater than 1)”

Authors’ response:

We fully agree with the reviewer concerning the need to invert HR as well as 95% CI. However, in the meantime, ECOG performance status and metastatic pattern at third-line initiation have been tested in a backward stepwise regression model (according to another reviewer’s suggestion) affecting selected covariates for multivariate analysis. The ECOG performance status (2-3 versus 0-1, HR: 4.04 [95% CI: 2.11-7.71], p<0.001) as well as tumor therapy (regorafenib versus TAS-102, HR: 1.95 [95% CI: 1.07-3.54], p=0.03) remained statistically significantly associated with hospitalization probability (for further details please see “point #10”). The latter findings have been updated in the “Abstract” and in the main text.

Reviewer #3 point #4:

In Table1 would be interesting to report and compare baseline characteristics stratified by therapy (i.e. regorafenib of TAS-102)“

Authors’ response:

According to the reviewer’s suggestion baseline characteristics were compared stratified by regorafenib and TAS-102 as third-line treatment in “Table 1” without detecting any statistically significant imbalances between patient groups.

Reviewer #3 point #5:

In Table2, please add percentage (%)”

Authors’ response:

With reference to reviewer’s point #1, we specified hospitalization outcome parameters in “Table 2” to “number of hospitalizations”, “number of hospitalized patients” including the percentages as well as to “range of hospitalizations per patient”.

Reviewer #3 point #6:

„Results: line 97: what do you mean with “median time of follow-up from third-line start”? Please define the follow-up period“

Authors’ response:

The median follow-up from third-line start was calculated using Kaplan-Meier curves where event indices (death versus censor) were switched. The definition has been added to the „Methods” section.

Reviewer #3 point #7:

Figure S2: the number of events seems to be too small for any statistical inference on the comparison of survival between groups“

Authors’ response:

The choice of third-line agent for mCRC treatment (regorafenib versus TAS-102) is still a matter of debate. Current ESMO and NCCN guidelines recommend both agents without favoring one drug over the other. Figure S2 aimed at providing the reader with real-world survival data according to sequencing strategy and monotherapy. Among 93 mCRC patients starting palliative third-line therapy with either regorafenib or TAS-102 death occurred in the majority of patients (56 out of 93; 60%).

Reviewer #3 point #8:

Figure S3: when comparing single-agent vs sequenced therapy, patients treated with sequenced therapy are affected by “immortal time” bias. Indeed, the time between initiation of third-line and initiation of fourth-line therapy is, by definition, “immortal”. This bias underestimates the survival of patients treated with sequenced therapy“

Authors’ response:

We totally agree with the reviewer on the issue of immortal time bias. In the initially submitted manuscript cross-over was taken into account as a time-dependent covariate in the Cox proportion hazard model in order to avoid the immortal time bias. We have added this sentence to the legend of Figure S3.

Reviewer #3 point #9:

„Similarly, Figure 3a/3b: in patients who experienced an hospitalization, the time from start of therapy to hospitalization is “immortal time” by definition, making the results biased“

Authors’ response:

In analogy to point #8, in the initially submitted manuscript, hospitalization was taken into account as a time-dependent covariate in the Cox proportion hazard model in order to avoid the immortal time bias. We have specified this sentence in the legend of Figure 2 (former Figure 3).

Reviewer #3 point #10:

Results: the backward stepwise regression selected only the treatment received (i.e. regorafenib or TAS-102) as covariate to be included in the final model. This may be due to the small number of patients included in the study cohort, which may be associated to a higher random variability (i.e. wide confidence intervals and low statistical power to detect significant associations). I found weird that age (for instance) is not associated to the risk of hospitalization. I would suggest to add to the model age and sex as covariates, even if not statistical significant“

Authors’ response:

Please note that according to other reviewers‘ suggestions the impact of peritoneal metastases as well as ECOG performance status (2-3 versus 0-1) at third-line initiation on hospitalization probability has been tested in a stepwise backward regression model with other covariates (age at third-line initiation, sex, therapy line, sidedness, primary tumor resection status, time point of metastases detection (synchronous versus metachronous), evidence of pulmonary metastases, evidence of liver metastases, RAS status, and tumor therapy (regorafenib versus TAS-102). The ECOG performance status, tumor therapy and sidedness were selected covariates for the final regression model. The ECOG performance status (2-3 versus 0-1, HR: 4.04 [95% CI: 2.11-7.71], p<0.001) as well as tumor therapy (regorafenib versus TAS-102, HR: 1.95 [95% CI: 1.07-3.54], p=0.03) remained statistically significantly associated with hospitalization probability, this was not the case for sidedness (right versus left, HR: 1.64 [95%CI: 0.91-2.95], p=0.10). We are confident that the ECOG performance status at initiation of third-line therapy more precisely reflects patients’ fitness compared to chronologic age. The abovementioned findings including multivariate analysis results in “Table 2” have been added to „Results“ section of the revised manuscript.

Multivariate analysis using covariates selected by a stepwise backward regression model (Table 2)

HR (95% CI)

p-value

tumor therapy (regorafenib versus TAS-102)

1.95 (1.07-3.54)

0.03

ECOG performance status at 3rd line start (2-3 versus 0-1)

4.04 (2.11-7.71)

<0.001

sidedness (right versus left)

1.64 (0.91-2.95)

0.1

According to the reviewer’s suggestion, we tested the impact of sex as well age (irrespective of statistical significance) in multivariate analysis (outcome: hospitalization) along with tumor therapy, ECOG performance status and sidedness (the latter three covariates were selected according to stepwise backward regression model). Only the ECOG performance status (2-3 versus 0-1, HR: 4.35 [95% CI: 2.12-8.92], p<0.001) and tumor therapy (regorafenib versus TAS-102, HR: 1.92 [95% CI: 1.06-3.50], p=0.031) remained statistically significantly associated with hospitalization probability (not included in the final manuscript).

Multivariate analysis (hospitalization probability) irrespective of statistical significance (selected covariates according to the reviewer’s suggestion, not included in the final manuscript):

HR (95% CI)

p-value

tumor therapy (regorafenib versus TAS-102)

1.92 (1.06, 3.5)

0.031

ECOG performance status (2-3 versus 0-1)

4.35 (2.12, 8.92)

<.001

sidedness (right versus left)

1.57 (0.82, 3.02)

0.18

age (<65 versus ≥65)

1.01 (0.54, 1.89)

0.98

sex (male versus female)

1.22 (0.64, 2.32)

0.54

Reviewer #3 point #11:

Figure 1: as the main outcome of interest is the hospitalization occurrence, I would suggest to delete the discontinuation without hospitalization curves from the figure, and to truncate the follow-up at 30 weeks, since no events occurred after that and given (I guess) the low number of patients at risk after 30 weeks.

Authors’ response:

We acknowledge the reviewer’s feedback on Figure 1. Due to the positive feedback on the competing risk analysis in Figure 1 by other reviewers in its current version and the additional information provided, we would prefer to keep Figure 1 in its current version (in case of agreement of the reviewer).

Reviewer #3 point #12:

“What does Figure 2 add to the results reported in Table 3?”

Authors’ response:

We agree with the reviewer about the redundancy of Figure 2 and Table 4 (former Table 3) and therefore removed Figure 2 from the manuscript.

Round 2

Reviewer 1 Report

The authors fully addressed our comments. I have no further improvement to suggest. 

Author Response

Point-by-point response to reviewer 1

Hospitalizations and Clinical Outcome in Metastatic Colorectal Cancer During Regorafenib or TAS-102 Therapy

cancers-912056

We thank reviewer 1 for reviewing our manuscript and the positive feedback concerning the requested revisions. All comments have been fully addressed. No further suggestions for improvement have been made.

Rievewer #1

„ The authors fully addressed our comments. I have no further improvement to suggest.“

Reviewer 3 Report

Point-by-point response to reviewer 3

Hospitalizations and Clinical Outcome in Metastatic Colorectal Cancer During Regorafenib or TAS-102 Therapy

cancers-912056

We thank reviewer 3 for the valuable input, which has definitely helped to improve our manuscript. The raised questions are answered in a point-by-point fashion and changes in the revised manuscript are highlighted via “track-change mode”.

Rievewer #3

“The current study aimed at evaluating and comparing hospitalization rates among metastatic colorectal cancer patients treated with either regorafenib or TAS-102 beyond second-line therapy. Moreover, the impact of hospitalizations of overall survival was also evaluated.

The study would be interesting from a clinical point of view, adding important information on the clinical impact of such therapies in the real-world clinical practice. However, I have several doubts and comments that I would like to be addressed. First of all, the following two important points, that would impact on the statistical analysis plan:”

Reviewer #3 point #1:

“In relation to the primary outcome (i.e. the occurrence of hospitalizations), it is not clear to me whether the outcome of interest is the first hospitalization occurred for each patients, or the total number of hospitalizations occurred during follow-up for each patient;“

Authors’ response:

We appreciate the reviewer’s question on this point. In the meanwhile, we have specified the outcome parameters in Table 2 and present outcome data according to therapy line as well as according to systemic therapy: 1) the total number of hospitalization (a single patient could be hospitalized several times), 2) the number of hospitalized patients (at least one hospitalization necessitated), 3) the percentage of hospitalized patients (≥1) as well as 4) the range of the number of hospitalizations per patient. The primary outcome in Figure 1 (competing risk analysis) is hospitalization probability, representing patients’ first hospitalization, assessing whether an individual patient was hospitalized or never-hospitalized during regorafenib versus TAS-102 treatment. Table 4 (former Table 3) shows the total number of hospitalizations and the respective hospitalization causes during third- and fourth-line therapy with regorafenib and TAS-102. Please note that an individual patient could have been hospitalized several times during regorafenib or TAS-102 therapy and as a consequence could “contribute” with multiple hospitalizations to Table 4.  Data presented in Table 5 depict whether any hospitalization in hospitalized patients (an individual patient could have been hospitalized several times) caused therapy discontinuation during regorafenib or TAS-102 treatment.

We made efforts to specify the respective outcome in the figure and table legends as well as in the text.

Comment: You should specify in the “Materials and Methods” section that the primary outcome of the study is the first hospitalization. Moreover, please specify which regression model you used for assessing the association between therapy and risk of hospitalization (I guess, a Cox proportional regression model).

Reviewer #3 point #2:

“It is unclear to me how you considered the exposure to regorafenib or TAS-102 during follow-up: did you employed an intention-to-treat design, by classifying patients as always exposed to the treatment received as third-line therapy, or you also considered switching from one therapy to another during follow-up? For example, in Figure 2, did you compare 69 and 24 patients treated, respectively, with third-line regorafenib or TAS-102, or you also included in the regorafenib group those patients who switched from TAS-102 to regorafenib?”

Authors’ response:

We appreciate the reviewer’s comment on this point. When comparing TAS-102 and regorafenib, switching from one therapy to another was always taken into account. In other words, in former Figure 2 (which has been removed according to the reviewer’s suggestions in the meanwhile due to redundancy with Table 4 (former Table 3)) all hospitalization causes during regorafenib (3rd line: n=69 patients + 4th line: n=7 patients) and TAS-102 (3rd line: n=24 patients + 4th line: n=31 patients) across 3rd and 4th line have been compared. Comparison of hospitalization causes during actual regorafenib and TAS-102 treatment period is shown according to separate treatment lines (3rd line as well as 4th line) as well as across 3rd line and 4th line in Table 4 (former Table 3).

Comment: It is still unclear to me how you considered person-time at risk for evaluating the risk of hospitalization with the Kaplan-Meier curves. If I understood well (but you should clearly define this statistical methods in the “method” section) you followed-up patients from start of third-line treatment until the date of hospitalization (if any) or the end of the study. What if a patient started third-line treatment with, say, regorafenib and then switched to TAS-102? How do you considered exposure for this patient? The same doubt regards the implementation of the regression model.

Reviewer #3 point #3:

“Abstract and Results: if stated that “a trend towards a higher hospitalization probability with regorafenib compared to TAS-102 (HR 0.62 [95% CI: 0.36 – 1.06])”, the HR should be inverted (i.e. the reference group should be TAS-102, and the HR should be greater than 1)”

Authors’ response:

We fully agree with the reviewer concerning the need to invert HR as well as 95% CI. However, in the meantime, ECOG performance status and metastatic pattern at third-line initiation have been tested in a backward stepwise regression model (according to another reviewer’s suggestion) affecting selected covariates for multivariate analysis. The ECOG performance status (2-3 versus 0-1, HR: 4.04 [95% CI: 2.11-7.71], p<0.001) as well as tumor therapy (regorafenib versus TAS-102, HR: 1.95 [95% CI: 1.07-3.54], p=0.03) remained statistically significantly associated with hospitalization probability (for further details please see “point #10”). The latter findings have been updated in the “Abstract” and in the main text.

Comment: Since the main objective of the study is the assessment of association between treatment and risk of hospitalization, in the Abstract please report first the HR related to treatment, and then that of ECOG.

Reviewer #3 point #4:

In Table1 would be interesting to report and compare baseline characteristics stratified by therapy (i.e. regorafenib of TAS-102)“

Authors’ response:

According to the reviewer’s suggestion baseline characteristics were compared stratified by regorafenib and TAS-102 as third-line treatment in “Table 1” without detecting any statistically significant imbalances between patient groups.

Reviewer #3 point #5:

In Table2, please add percentage (%)”

Authors’ response:

With reference to reviewer’s point #1, we specified hospitalization outcome parameters in “Table 2” to “number of hospitalizations”, “number of hospitalized patients” including the percentages as well as to “range of hospitalizations per patient”.

Reviewer #3 point #6:

„Results: line 97: what do you mean with “median time of follow-up from third-line start”? Please define the follow-up period“

Authors’ response:

The median follow-up from third-line start was calculated using Kaplan-Meier curves where event indices (death versus censor) were switched. The definition has been added to the „Methods” section.

Comment: I would like to clearly see in the “Method” section how do you considered patient’s follow-up: from which date to which date you followed-up patients? Moreover, it is unclear to me the meaning of “event indices (death versus censor) were switched”.

Reviewer #3 point #7:

Figure S2: the number of events seems to be too small for any statistical inference on the comparison of survival between groups“

Authors’ response:

The choice of third-line agent for mCRC treatment (regorafenib versus TAS-102) is still a matter of debate. Current ESMO and NCCN guidelines recommend both agents without favoring one drug over the other. Figure S2 aimed at providing the reader with real-world survival data according to sequencing strategy and monotherapy. Among 93 mCRC patients starting palliative third-line therapy with either regorafenib or TAS-102 death occurred in the majority of patients (56 out of 93; 60%).

Reviewer #3 point #8:

Figure S3: when comparing single-agent vs sequenced therapy, patients treated with sequenced therapy are affected by “immortal time” bias. Indeed, the time between initiation of third-line and initiation of fourth-line therapy is, by definition, “immortal”. This bias underestimates the survival of patients treated with sequenced therapy“

Authors’ response:

We totally agree with the reviewer on the issue of immortal time bias. In the initially submitted manuscript cross-over was taken into account as a time-dependent covariate in the Cox proportion hazard model in order to avoid the immortal time bias. We have added this sentence to the legend of Figure S3.

Comment: please specify the use of the time-dependent variable in the “Method” section. Moreover, did you use a time-dependent covariate in the Cox model or in the Kaplan-Meier curves? This is unclear.

Reviewer #3 point #9:

„Similarly, Figure 3a/3b: in patients who experienced an hospitalization, the time from start of therapy to hospitalization is “immortal time” by definition, making the results biased“

Authors’ response:

In analogy to point #8, in the initially submitted manuscript, hospitalization was taken into account as a time-dependent covariate in the Cox proportion hazard model in order to avoid the immortal time bias. We have specified this sentence in the legend of Figure 2 (former Figure 3).

Comment: please see the comment above.

Reviewer #3 point #10:

Results: the backward stepwise regression selected only the treatment received (i.e. regorafenib or TAS-102) as covariate to be included in the final model. This may be due to the small number of patients included in the study cohort, which may be associated to a higher random variability (i.e. wide confidence intervals and low statistical power to detect significant associations). I found weird that age (for instance) is not associated to the risk of hospitalization. I would suggest to add to the model age and sex as covariates, even if not statistical significant“

Authors’ response:

Please note that according to other reviewers‘ suggestions the impact of peritoneal metastases as well as ECOG performance status (2-3 versus 0-1) at third-line initiation on hospitalization probability has been tested in a stepwise backward regression model with other covariates (age at third-line initiation, sex, therapy line, sidedness, primary tumor resection status, time point of metastases detection (synchronous versus metachronous), evidence of pulmonary metastases, evidence of liver metastases, RAS status, and tumor therapy (regorafenib versus TAS-102). The ECOG performance status, tumor therapy and sidedness were selected covariates for the final regression model. The ECOG performance status (2-3 versus 0-1, HR: 4.04 [95% CI: 2.11-7.71], p<0.001) as well as tumor therapy (regorafenib versus TAS-102, HR: 1.95 [95% CI: 1.07-3.54], p=0.03) remained statistically significantly associated with hospitalization probability, this was not the case for sidedness (right versus left, HR: 1.64 [95%CI: 0.91-2.95], p=0.10). We are confident that the ECOG performance status at initiation of third-line therapy more precisely reflects patients’ fitness compared to chronologic age. The abovementioned findings including multivariate analysis results in “Table 2” have been added to „Results“ section of the revised manuscript.

Multivariate analysis using covariates selected by a stepwise backward regression model (Table 2)

HR (95% CI)

p-value

tumor therapy (regorafenib versus TAS-102)

1.95 (1.07-3.54)

0.03

ECOG performance status at 3rd line start (2-3 versus 0-1)

4.04 (2.11-7.71)

<0.001

sidedness (right versus left)

1.64 (0.91-2.95)

0.1

According to the reviewer’s suggestion, we tested the impact of sex as well age (irrespective of statistical significance) in multivariate analysis (outcome: hospitalization) along with tumor therapy, ECOG performance status and sidedness (the latter three covariates were selected according to stepwise backward regression model). Only the ECOG performance status (2-3 versus 0-1, HR: 4.35 [95% CI: 2.12-8.92], p<0.001) and tumor therapy (regorafenib versus TAS-102, HR: 1.92 [95% CI: 1.06-3.50], p=0.031) remained statistically significantly associated with hospitalization probability (not included in the final manuscript).

Multivariate analysis (hospitalization probability) irrespective of statistical significance (selected covariates according to the reviewer’s suggestion, not included in the final manuscript):

HR (95% CI)

p-value

tumor therapy (regorafenib versus TAS-102)

1.92 (1.06, 3.5)

0.031

ECOG performance status (2-3 versus 0-1)

4.35 (2.12, 8.92)

<.001

sidedness (right versus left)

1.57 (0.82, 3.02)

0.18

age (<65 versus ≥65)

1.01 (0.54, 1.89)

0.98

sex (male versus female)

1.22 (0.64, 2.32)

0.54

Reviewer #3 point #11:

Figure 1: as the main outcome of interest is the hospitalization occurrence, I would suggest to delete the discontinuation without hospitalization curves from the figure, and to truncate the follow-up at 30 weeks, since no events occurred after that and given (I guess) the low number of patients at risk after 30 weeks.

Authors’ response:

We acknowledge the reviewer’s feedback on Figure 1. Due to the positive feedback on the competing risk analysis in Figure 1 by other reviewers in its current version and the additional information provided, we would prefer to keep Figure 1 in its current version (in case of agreement of the reviewer).

Reviewer #3 point #12:

“What does Figure 2 add to the results reported in Table 3?”

Authors’ response:

We agree with the reviewer about the redundancy of Figure 2 and Table 4 (former Table 3) and therefore removed Figure 2 from the manuscript.

Author Response

Point-by-point response to reviewer 3

(Revision 2)

Hospitalizations and Clinical Outcome in Metastatic Colorectal Cancer During Regorafenib or TAS-102 Therapy

cancers-912056

Changes in the manuscript (from revision 1 to revision 2) have been highlighted in yellow in the manuscript.

Reviewer #3 point #1:

“In relation to the primary outcome (i.e. the occurrence of hospitalizations), it is not clear to me whether the outcome of interest is the first hospitalization occurred for each patients, or the total number of hospitalizations occurred during follow-up for each patient;“

Authors’ response (round 1):

We appreciate the reviewer’s question on this point. In the meanwhile, we have specified the outcome parameters in Table 2 and present outcome data according to therapy line as well as according to systemic therapy: 1) the total number of hospitalization (a single patient could be hospitalized several times), 2) the number of hospitalized patients (at least one hospitalization necessitated), 3) the percentage of hospitalized patients (≥1) as well as 4) the range of the number of hospitalizations per patient. The primary outcome in Figure 1 (competing risk analysis) is hospitalization probability, representing patients’ first hospitalization, assessing whether an individual patient was hospitalized or never-hospitalized during regorafenib versus TAS-102 treatment. Table 4 (former Table 3) shows the total number of hospitalizations and the respective hospitalization causes during third- and fourth-line therapy with regorafenib and TAS-102. Please note that an individual patient could have been hospitalized several times during regorafenib or TAS-102 therapy and as a consequence could “contribute” with multiple hospitalizations to Table 4.  Data presented in Table 5 depict whether any hospitalization in hospitalized patients (an individual patient could have been hospitalized several times) caused therapy discontinuation during regorafenib or TAS-102 treatment.

We made efforts to specify the respective outcome in the figure and table legends as well as in the text.

Comment: You should specify in the “Materials and Methods” section that the primary outcome of the study is the first hospitalization. Moreover, please specify which regression model you used for assessing the association between therapy and risk of hospitalization (I guess, a Cox proportional regression model).

Authors’ response (round 2):

The primary outcome of our analysis was cumulative risk for hospitalization during treatment with regorafenib and TAS-102 as third- or fourth-line therapy, respectively. The first hospitalization with a hospital admission for at least one night was the event of interest. Multivariate analysis was based on a Fine-Gray proportional subdistribution hazards regression model. This specification has been added to the “Materials and Methods” section.

Reviewer #3 point #2:

“It is unclear to me how you considered the exposure to regorafenib or TAS-102 during follow-up: did you employed an intention-to-treat design, by classifying patients as always exposed to the treatment received as third-line therapy, or you also considered switching from one therapy to another during follow-up? For example, in Figure 2, did you compare 69 and 24 patients treated, respectively, with third-line regorafenib or TAS-102, or you also included in the regorafenib group those patients who switched from TAS-102 to regorafenib?”

Authors’ response (round 1):

We appreciate the reviewer’s comment on this point. When comparing TAS-102 and regorafenib, switching from one therapy to another was always taken into account. In other words, in former Figure 2 (which has been removed according to the reviewer’s suggestions in the meanwhile due to redundancy with Table 4 (former Table 3)) all hospitalization causes during regorafenib (3rdline: n=69 patients + 4th line: n=7 patients) and TAS-102 (3rd line: n=24 patients + 4th line: n=31 patients) across 3rd and 4th line have been compared. Comparison of hospitalization causes during actual regorafenib and TAS-102 treatment period is shown according to separate treatment lines (3rd line as well as 4th line) as well as across 3rd line and 4th line in Table 4 (former Table 3).

Comment: It is still unclear to me how you considered person-time at risk for evaluating the risk of hospitalization with the Kaplan-Meier curves. If I understood well (but you should clearly define this statistical methods in the “method” section) you followed-up patients from start of third-line treatment until the date of hospitalization (if any) or the end of the study. What if a patient started third-line treatment with, say, regorafenib and then switched to TAS-102? How do you considered exposure for this patient? The same doubt regards the implementation of the regression model.

Authors’ response (round 2):

The first hospitalization with a hospital admission for at least one night during treatment with TAS-102 or regorafenib was the event of interest. If a patient was treated sequentially with both mentioned drugs, the patient was considered twice for this analysis, for the third- and fourth-line treatment with the respective drug. Multivariate analysis was based on a Fine-Gray proportional subdistribution hazards regression model. Both, treatment line, as well as the administered drug (TAS-102 or regorafenib) were covariates included in the backward stepwise regression model. These specifications have been added to the “Materials and Methods” section.

Reviewer #3 point #3:

“Abstract and Results: if stated that “a trend towards a higher hospitalization probability with regorafenib compared to TAS-102 (HR 0.62 [95% CI: 0.36 – 1.06])”, the HR should be inverted (i.e. the reference group should be TAS-102, and the HR should be greater than 1)”

Authors’ response (round 1):

We fully agree with the reviewer concerning the need to invert HR as well as 95% CI. However, in the meantime, ECOG performance status and metastatic pattern at third-line initiation have been tested in a backward stepwise regression model (according to another reviewer’s suggestion) affecting selected covariates for multivariate analysis. The ECOG performance status (2-3 versus0-1, HR: 4.04 [95% CI: 2.11-7.71], p<0.001) as well as tumor therapy (regorafenib versus TAS-102, HR: 1.95 [95% CI: 1.07-3.54], p=0.03) remained statistically significantly associated with hospitalization probability (for further details please see “point #10”). The latter findings have been updated in the “Abstract” and in the main text.

Comment: Since the main objective of the study is the assessment of association between treatment and risk of hospitalization, in the Abstract please report first the HR related to treatment, and then that of ECOG.

Authors’ response (round 2):

According to the reviewer’s suggestion, the HR related to treatment (regorafenib versus TAS-102) has been reported first in the abstract.

Reviewer #3 point #6:

„Results: line 97: what do you mean with “median time of follow-up from third-line start”? Please define the follow-up period“

Authors’ response (round 1):

The median follow-up from third-line start was calculated using Kaplan-Meier curves where event indices (death versus censor) were switched. The definition has been added to the „Methods” section.

Comment: I would like to clearly see in the “Method” section how do you considered patient’s follow-up: from which date to which date you followed-up patients? Moreover, it is unclear to me the meaning of “event indices (death versus censor) were switched”.

Authors’ response (round 2):

For survival analysis, median follow-up from initiation of third-line treatment with either regorafenib or TAS-102 was calculated using the Kaplan-Meier estimator with reversed status indicators (death and censored). This has been clarified in the Methods section.

Reviewer #3 point #8:

“Figure S3: when comparing single-agent vs sequenced therapy, patients treated with sequenced therapy are affected by “immortal time” bias. Indeed, the time between initiation of third-line and initiation of fourth-line therapy is, by definition, “immortal”. This bias underestimates the survival of patients treated with sequenced therapy“

Authors’ response (round 1):

We totally agree with the reviewer on the issue of immortal time bias. In the initially submitted manuscript cross-over was taken into account as a time-dependent covariate in the Cox proportion hazard model in order to avoid the immortal time bias. We have added this sentence to the legend of Figure S3.

Comment: please specify the use of the time-dependent variable in the “Method” section. Moreover, did you use a time-dependent covariate in the Cox model or in the Kaplan-Meier curves? This is unclear.

Authors’ response (round 2):

For survival analysis according to treatment groups, adjusted survival curves using Cox proportional hazards model were created. To avoid immortal time bias, cross-over was taken into account as a time-dependent covariate beginning with the start of fourth-line treatment. This clarification has been added to the methods section and to the figure legend.

Reviewer #3 point #9:

„Similarly, Figure 3a/3b: in patients who experienced an hospitalization, the time from start of therapy to hospitalization is “immortal time” by definition, making the results biased“

Authors’ response (round 1):

In analogy to point #8, in the initially submitted manuscript, hospitalization was taken into account as a time-dependent covariate in the Cox proportion hazard model in order to avoid the immortal time bias. We have specified this sentence in the legend of Figure 2 (former Figure 3).

Comment: please see the comment above.

Authors’ response (round 2):

Please see response to comment #8.
